# Delivery of Melittin as a Lytic Agent via Graphene Nanoparticles as Carriers to Breast Cancer Cells

**DOI:** 10.3390/jfb13040278

**Published:** 2022-12-07

**Authors:** Karolina Daniluk, Agata Lange, Michał Pruchniewski, Artur Małolepszy, Ewa Sawosz, Sławomir Jaworski

**Affiliations:** 1Department of Nanobiotechnology, Institute of Biology, Warsaw University of Life Sciences, 02-786 Warsaw, Poland; 2Faculty of Chemical and Process Engineering, Warsaw University of Technology, 00-654 Warsaw, Poland

**Keywords:** melittin, internalization, drug delivery, carbon nanoparticles

## Abstract

Melittin, as an agent to lyse biological membranes, may be a promising therapeutic agent in the treatment of cancer. However, because of its nonspecific actions, there is a need to use a delivery method. The conducted research determined whether carbon nanoparticles, such as graphene and graphene oxide, could be carriers for melittin to breast cancer cells. The studies included the analysis of intracellular pH, the potential of cell membranes, the type of cellular transport, and the expression of receptor proteins. By measuring the particle size, zeta potential, and FT-IT analysis, we found that the investigated nanoparticles are connected by electrostatic interactions. The level of melittin encapsulation with graphene was 86%, while with graphene oxide it was 78%. A decrease in pHi was observed for all cell lines after administration of melittin and its complex with graphene. The decrease in membrane polarization was demonstrated for all lines treated with melittin and its complex with graphene and after exposure to the complex of melittin with graphene oxide for the MDA-MB-231 and HFFF2 lines. The results showed that the investigated melittin complexes and the melittin itself act differently on different cell lines (MDA-MB-231 and MCF-7). It has been shown that in MDA-MD-231 cells, melittin in a complex with graphene is transported to cells via caveolin-dependent endocytosis. On the other hand, the melittin–graphene oxide complex can reach breast cancer cells through various types of transport. Other differences in protein expression changes were also observed for tumor lines after exposure to melittin and complexes.

## 1. Introduction

Breast cancer is the most widely occurring cancer in women and the most prevalent cancer overall [1]. According to the World Health Organization, it accounted for 12% of all cancer cases in 2021. However, prophylactic programs for this type of cancer introduced in many countries allow for early diagnosis and implementation of early treatment, which contributed to a decrease in mortality by 1% annually in 2013–2018 [2]. Despite developments in medicine and the growing interest in new methods of cancer treatment, according to the American Cancer Society, it is estimated that 1 in 39 women will die of breast cancer in 2022. 

Products from the European honeybee (*Apis mellifera*), such as honey, propolis, and venom, are natural elements in the treatment of many diseases [3,4,5,6,7,8,9]. The main component in bee venom is melittin (MEL), which accounts for about half of the dry weight of the venom [10]. It is a 26-amino-acid peptide of amphiphilic nature, which means that it is soluble in both polar, such as water, and nonpolar solvents, such as toluene [11]. 

Because of its structure, MEL is capable of biological interaction with cell membranes. It binds to the phospholipids in the lipid bilayer and forms tetramers, which create pores in membranes with a diameter of about 4.4 nm [12]. The resulting pores disturb the integrity of the cell membranes, which disrupts the internal balance of cells, leading to their damage and death [13].

Because of the mechanism of its interaction with biological structures, MEL has been the subject of many studies in microbiology, veterinary sciences, and medicine [14,15,16]. MEL has an affinity for the negative charge of the envelope of bacterial cells, which leads to a disturbance in this structure. Additionally, its antibacterial properties were demonstrated against many groups of bacteria, including *Pseudomonas aeruginosa*, multidrug-resistant *Acinetobacter baumanni*, *E. coli*, and *Staphylococcus aureus* [17,18,19]. The main purpose of this peptide is perforation of the bacterial cell wall. In addition to interacting with the external structures of bacteria, MEL can also penetrate inside and damage proteins, DNA, and RNA, disrupting intracellular processes [20].

Bee venom has been used extensively in conventional medicine for many years and is now being studied for its action in a variety of diseases [21,22,23]. The main mechanism of MEL is membrane action, which contributes to the increased interest in its role in clinical application in anticancer research. MEL’s cytotoxic effect on many cell lines has been demonstrated many times. It has been proven that it causes apoptosis, i.e., natural cell death, e.g., in human leukemia and prostate cells [24,25]. However, in hepatocellular carcinoma cells, MEL induces necrosis, and in gastric cancer, apoptosis and necrosis [26,27,28]. In our previous studies, it has been shown that MEL causes apoptosis and necrosis in adenocarcinoma breast cancer cells [29]. Moreover, it influences the migration and mobility of cells, angiogenesis, and proliferation of neoplastic cells [30,31]. Therefore, it is important to study its interactions separately for each type of cancer cell. However, studies showed that MEL also works on healthy cells, which makes it possible to look for ways to deliver it to the cells that we want to eliminate [32,33,34].

In recent years, many nanomaterials have been extensively studied as a drug delivery system for cancer cells [35]. Many studies indicated the potential use of graphene nanomaterials as MEL carriers because of the mechanism of uptake into cells’ MEL delivery. It was proven that graphene (GN) and graphene oxide (GO) move inside cells by internalization, which consists of transporting particles through the invagination of the membrane [36]. Furthermore, GN-based nanomaterials can be inhibitors of invasion and migration of breast cancer cells and limit the metastasis and growth of tumors by inhibiting mitochondrial respiration [37]. It was shown that after exposure to GO, the permeability of the cell membrane increased, which indicates its potential use as a carrier for MEL [38]. Using the mechanism of internalization of graphene nanoparticles, which has been proven many times, we assume that MEL binding to them will be transported inside cancer cells, which will cause its effect on intracellular membrane structures [39,40]. Such MEL delivery may cause disintegration of intracellular membranes and activation of natural cell death [40,41]. GN and GO can be carriers for MEL and will have a stronger toxicity effect inside cancer cells. The aim of this study was to evaluate the transport type of MEL complexes with GN and GO to breast cancer cells.

## 2. Materials and Methods

### 2.1. Complex Preparation and Characterization

Pure MEL peptide was obtained from Sigma-Aldrich (purity ≥ 85%; Munich, Germany) in powder form and dissolved in ultrapure water. GN (thickness: 6–8 nm; purity > 99.5%) powder was purchased from SkySpring Nanomaterials (Houston, TX, USA) and GO (thickness < 2 nm; purity 98%; diameter: 8–15 um) powder was purchased from the Institute of Electronic Materials Technology in Warsaw, Poland. MEL was added to each type of nanoparticle to obtain two different complexes in concentrations of 20 μg/mL for nanoparticles and 10 μg/mL for MEL. To obtain complexes by self-organization, the complexes were incubated at room temperature and vortexed for 15 min. Size distribution (dynamic light scattering method) and Zeta potential (laser Doppler electrophoresis method) of MEL, GN, GO, and colloid complexes were measured using a Zetasizer Nano ZS, model ZEN3600 (Malvern Instruments, Malvern, UK). Concentrations of 10 μg /mL MEL and 20 μg /mL for GN and GO were used for the analysis. Each sample was measured after 120 s of stabilization at 25 °C. The measurements of the complexes were measured immediately after 15 min of vortexing. Fourier-transform infrared (FT-IR) spectroscopy measurements were performed on a Nicolet iS10 (Thermo Scientific) spectrometer. Before the sample measurement, “dry air” background was recorded, which was subtracted automatically during the registration of spectra for the investigated samples. The samples were mixed with KBr at a ratio of 1/300 mg and then compressed at 7 MPa cm^−2^ to form a pellet and the transmission spectrum was recorded. The spectra were collected in a range 400–4000 cm^−1^.

### 2.2. Entrapment Efficiency (EE%) of the MEL in Complexes

The complexes were obtained from the determination of free MEL concentration in supernatant recovered from the centrifugation process. The concentrations of MEL were measured using a NanoDrop 2000 spectrophotometer (Thermo Scientific, Wilmington, DE, USA) at 280 nm using absorption coefficient of 5570 M^−1^ cm^−1^ [42]. Concentrations of 10 μg /mL MEL and 20 μg /mL for GN and GO were used for the analysis. Encapsulation efficiency was estimated using the following equation:Encapsulation efficiency (%) = [ (A − B)/A ] × 100
“A” is the amount of initial MEL trapped in the nanoparticles and “B” is the unloaded MEL. 

### 2.3. Cell Culture

Human breast adenocarcinoma MCF-7, MDA-MB-231, and human fetal foreskin fibroblast HFFF2 cell lines were obtained from American Type Culture Collection (Manassas, VA, USA) and maintained in Dulbecco’s modified Eagle’s culture medium containing 10% fetal bovine serum (Life Technologies, Houston, TX, USA), 1% penicillin, and streptomycin (Life Technologies) at 37 °C in a humidified atmosphere of 5% CO_2_/95% air in a NuAireDH AutoFlow CO_2_ air-jacketed incubator (Plymouth, MN, USA).

### 2.4. Confocal Microscopy

The changes in the structure of the cell skeleton and nucleus were evaluated by confocal microscopy (Olympus FV1000, Tokyo, Japan). Cells were grown on a coverslip in a 6-well plate for 4 days and then the test reagents were added. After 24 h of incubation with MEL, complex cells were fixed with 4% paraformaldehyde in PBS (Life Technologies, Texas, TX, USA) for 10 min at room temperature and washed with ice-cold PBS. The next step was incubation in Triton X-100 (Sigma-Aldrich, Missouri, MO, USA) and washing with PBS (Sigma-Aldrich, Missouri, MO, USA). Then, the cells were incubated with normal goat serum as a blocking solution (Chemicon International, California, CA, USA). Actin filaments were stained with phalloidin-Atto 633 (Sigma-Aldrich, Missouri, MO, USA) and cell nuclei with 4′,6-diamidino-2-phenylindole (DAPI) (Thermo Fisher Scientific, Massachusetts, MA, USA). Each group was prepared in 5 replications. The photos were taken in 7 randomly selected fragments.

### 2.5. Intracellular pH

Intracellular pH (pHi) measurements were carried out using the Fluorometric Intracellular pH Assay Kit (MAK150, Sigma) according to the manufacturer’s instructions. MDA-MB-231, MCF-7, and HFFF2 cells were seeded into 96-well plates (40,000 cells/well) in culture media. After 24 h of incubation, the growth medium was replaced with 100 µL of HBBS buffer. Then, dye loading solution was added (100 µL/well) and incubated protected from light in a 5% CO_2_/95% air, 37 °C incubator for 30 min. Subsequently, a solution of MEL, nanoparticles, and complexes in HBBS buffer was added (50 μL/well) to obtain final concentrations 20 μg/mL for nanoparticles and 10 μg/mL for MEL. Immediately after adding the compounds, fluorescence was measured at λ_ex_ = 490/λ_em_ = 535 nm (cut off at 515 nm) using a Tecan Infinite 200 microplate reader (Tecan, Durham, NC, USA).

### 2.6. Potential of Cell Membrane

The Cellular Membrane Potential Assay Kit (ab176764, Abcam, Cambridge, UK) was used to study changes in cell membrane potential. Cells were plated on black 96-well plates at approximately 3 × 10^4^ cells per well and incubated for 24 h. Then, the medium was replaced with the solutions of MEL, GN, GO, and complexes (in which the concentration of MEL was 10 µg/mL and for nanoparticles 20 µg/mL) in the medium. After 4 h of incubation, the solutions were removed and 100 µL of assay buffer with 1.5 µL of MP sensor dye was added. After 30 min incubation at room temperature in the dark, the level of fluorescence was measured at Ex/Em = 530/570 nm with an enzyme-linked immunosorbent assay (ELISA) reader.

### 2.7. Transport Inhibitor Test

To investigate the mechanism of transport to the cells of the tested components, a viability test was performed after exposure to inhibitors of various types of transport using Cell Proliferation Kit II (Cat. No. 11465015001, Merck, Darmstadt, Germany). The following transport inhibitors were selected for the study: Pitstop-2 (5 µM), colchicine (10 µg/mL), cytochalasin D (5 µg/mL), genistein (200µM), phenylarsine oxide (5 μM), chlorpromazine (10 µg/mL), and dynasore hydrate (250 µM). MDA-MB-231, MCF-7, and HFFF2 cells were seeded into 96-well plates at 10^4^ cells/100 µL/well in culture media. After 24 h of incubation, the growth medium was replaced with 100 μL of inhibitor solutions and prepared in the culture medium. After 120 min of incubation with transport inhibitors, 50 μL of MEL, nanoparticles, and complexes was added to obtain final concentrations 20 μg/mL for nanoparticles and 10 μg/mL for MEL separately and in complexes. After 120 min of incubation, XTT assay was performed according to the manufacturer’s instructions. After 4 h of incubation, absorbance at 450 nm was measured using a microplate Elisa reader (Infinite M200, Tecan, Durham, NC, USA). The results were repeated five times for each group. Cell viability was expressed as a percentage of the optical density of the test sample reduced by a blank probe in relation to the optical density of the control reduced by a blank probe, where the control is the optical density of the cells without nanoparticles and the blank probe is optical density of the wells without cells.

### 2.8. Human Receptor Antibody Array

Adenocarcinoma breast cancer MDA-MB-231 and MCF-7 cell lines were treated with MEL (10 μg/mL) and complexes MEL with GN (MELGN), MEL with GO (MELGO) (MEL concentration 10 μg/mL in complexes, GO, GN—20 μg/mL concentration in complexes) and incubated for 24 h. The cells were scraped off, centrifuged, and washed twice in PBS. Cells not treated with MEL or complexes were used as a control. The cell pellet was resuspended in a diluted lysis buffer containing protease and phosphatase inhibitors (Sigma-Aldrich, St. Louis, MO, USA) according to the manufacturer’s instructions. Frozen metal balls and TissueLyser (Qiagen, Hilden, Germany) were used for homogenization at 50 Hz for 10 min on a shaking frozen cartridge. The samples were then centrifuged (30 min; 14,000× *g*; 4 °C), and the supernatant was collected. Protein concentration was determined using a bicinchoninic acid kit (Sigma-Aldrich, St. Louis, MO, USA). Analysis of receptor cell membranes was performed using an antibody array (ab211065; Abcam, Cambridge, UK). The assay was performed in accordance with the manufacturer’s instructions, using lysates containing 500 μg/mL of total protein per membrane. Membranes were visualized using the Azure Biosystem C400 (Azure, Dublin, CA, USA) [43]. Membrane photos were analyzed in imageJ using Protein Array Analyzer. Results were normalized and compared to a dots control sample.

### 2.9. Statistical Analysis

For this research, all data were represented as mean ± standard deviation. For statistical analysis, one-way analysis of variance with the post hoc Tukey test (HSD) was performed using GraphPad Prism 9 software, and the significance level was considered at *p*-value ≤ 0.05.

## 3. Results

### 3.1. Entrapment Efficiency

In the analysis using UV-VIS technology, EE% = 86% for the MEL complex with GN and 78% for the MEL complex with GO were noted. Values of EE% > 80 are considered good, but the result for MGO was close to this value.

### 3.2. Characterization of Complexes

DLS analyses showed the presence of GN structures in a range of 300–500 nm and, for GO, a bimodal distribution in a range of 200 to 2000 nm and 3000 to 7000 nm, which may be the result of agglomeration of GO structures (Figure 1). Zeta potential analysis showed a positive charge for the MEL alone and a negative charge for the nanoparticles alone (Figure 2).

The FT-IR spectra of GN, GO, MEL, and their complexes are shown in Figure 3. The broad peak observed between 3000 and 3700 cm^−1^ is assigned mainly to water and hydroxyl groups (O–H)ν. The broadband at 3400–3300 cm^−1^ could be assigned to N–H stretching vibration in MEL. Smaller features from 2850 to about 3000 cm^−1^ can be attributed to the C–H stretch. The peak at 1650 cm^−1^ can be assigned to C=O stretching vibrations of GO or C=N stretching vibrations of MEL and MEL complexes. The peak around 1610 cm^−1^ can be assigned to aromatic (sp2 vibrational) C=C bonds present in graphitic carbon. Other peaks observed around 1500 cm^−1^ on the FT-IR spectrum represent the N–H bending vibration of NH_2_, and C–O stretching vibrations from the C-terminal amino acid are assigned at 1050–1150 cm^−1^ [44,45].

### 3.3. Confocal Microscopy

The confocal microscope imaging of the actin filament structure and nucleus is shown in Figure 4 and Figure 5. Cell shrinkage of all lines was observed after MEL treatment. The tested MGN and MGO complexes resulted in a reduction in cell nuclei and a large shrinkage of cell bodies, and a decrease in the number of cells was observed for all three tested lines: MDA-MB-231, MCF-7, and HFFF2.

### 3.4. Intracellular pH

The results of the intracellular pH test are shown in Figure 6 for the three tested cell lines. An increase in fluorescence was observed for the group treated with MEL alone and with the MGN complex compared to the control group in all cell lines.

### 3.5. Potential of Cell Membrane

The results of the cell membrane potential analysis are shown in Figure 7. After 4 h, significant differences were obtained in the potential of cell membranes for all lines in individual research groups: for the MDA-MB-231 line in M, MGN, and MGO groups (respectively, decrease: 882, 1106, 551 relative fluorescence units (RFUs)); for the MCF-line in the M group and MGN (respectively, decrease: 929, 847 RFUs); and for the HFFF2 line in the M, MGN, and MGO groups (respectively, decrease: 384, 370, 356 RFUs).

### 3.6. Transport Inhibitor Test

The conducted viability test in response to MEL, nanoparticles, and MGN, MGO complexes, with the simultaneous use of transport blockers, made it possible to examine whether the materials are taken up by particular types of cellular transport. The results are shown in Figure 8, Figure 9 and Figure 10. It has been shown that individual lines can transport the tested materials in a different way.

### 3.7. Human Receptor Antibody Array

The protein expression of the extracellular receptor was tested using Human Receptor Antibody Array. A total of 40 proteins was analyzed: 4-1BB (TNFRSF9), ALCAM (CD166), CD80, BCMA, CD14, CD30, CD40 Ligand, CEACAM-1, DR6, Dtk, Endoglin, ErbB3, E-Selectin, Fas, Flt-3 Ligand, GITR, HVEM, ICAM-3, IL-1 R4, IL-1 R1, IL10 Rbeta, IL-17RA, IL-1 R gamma, IL-21R, LIMPII, Lipocalin-2 (NGAL), L-Selectin, LYVE-1, MICA, MICB, NRG1-beta 1, PDGF R beta, PECAM-1, RAGE, TIM-1, TRAIL R3 (TNFRSF10C), Trappin-2, uPar, VCAM-1, and XEDAR. The results are shown in Figure 11 and Figure 12. An increase in expression was observed for the TRAIL R3 protein in MEL-treated and Dtk in MGN-treated MCF-7 cells. Moreover, in line with MDA-MB-231, an increase in expression of the following proteins was observed: 4-1BB (MEL-treated group), ALCAM (MGN-treated group), and uPar (MGN- and MGO-treated group). A decrease in expression was observed for MICA and ALCAM in MGN- and MGO-treated MCF-7 cells and Lipo-2 (MGN- and MGO-treated group) and MICA (M-treated group) in MDA-MB-231 cells.

## 4. Discussion

For decades, natural ingredients have been employed for therapeutic purposes in terms of their beneficial effects in easing and treating a variety of illnesses, including cancer. Breast cancer is the second-most-common cause of cancer-related death for American women and the primary cause of cancer death for women globally. Breast cancer risk can be determined using risk assessment methods, and patients who are at a high risk may be candidates for drugs that lower their risk [46,47].

MEL is a bee venom compound with lytic properties against many types of cells [48]. In addition, it is cell nonspecific and exhibits hemolytic properties, which are a challenge for the use of this peptide in medicine [49]. Researchers tested MEL for its anticancer properties against a number of cancers, including breast cancer [50,51,52]. The use of nanoliposomes and MEL encapsulation with polomaxer 188 has been shown to control hepatocellular carcinoma tumors without serious side effects [53]. In turn, the use of citric-acid-functionalized Fe_3_O_4_ magnetic nanoparticles increased the toxic effect, but the research was carried out only on one cell line, and the negative aspects of the effect of the tested complexes were not taken into account [54]. Various methods of delivering MEL to breast cancer cells have already been tested [55,56]. One of them was niosomes, where it was shown that their use increases the toxicity effect of MEL itself, but an equally strong toxic effect was also observed on a noncancer cell line [57]. So far, there are no broader reports on the use of GN and GO as MEL carriers. There are reports on the application of inorganic, lipid-based, and polymeric MEL carriers; therefore, our results complement the gaps in the knowledge on the use of GN and GO as MEL carriers, especially for breast cancer cells [54,58,59,60]. GO in the PEG-GO-Fe_3_O_4_ complex has been studied as one of the elements in the delivery system to HeLa cells [61]. Studies have shown that GO acts as an MEL stabilizer; however, the entire complex causes the lysis of cancer cells. In our previous study, it was shown that complexes have a stronger toxic effect on breast cancer cells than MEL alone, especially for the line MDA-MB-231 [29]. Moreover, nanodiamond inhibited the lytic effects of MEL; therefore, it was not studied in the subsequent parts of the studies described in this paper. It has been proven that the complexes induce a lower level of necrosis in cells compared to MEL. In this study, we used two carbon nanoparticles, GN and GO, as MEL carriers to reduce the effect of MEL on outer cell membranes via the uptake of complexes with nanoparticles by cells, which will allow for the lytic effect of MEL on intracellular membrane structures in breast cancer cells. 

The DLS results of the MGO sample showed that the use of MEL affects the stabilization of GO particles in the colloids and prevents their aggregation. Further, the average DLS values are higher for the MGN and GO complexes than the nanoparticles themselves, which may indicate the formation of complexes of the nanoparticles with MEL. Since the FTIR spectra do not show new or more intense bands, it can be concluded that MEL and carbon nanoparticles are connected by a physical bond (Figure 13). The results of the Zeta potential showed the opposite charge of the components, which confirms the electrostatic interaction occurring in the complexes [62].

Analysis of the visualization of cell nuclei and actin fibers using a confocal microscope slowly observed differences in all tested lines between the research groups. For all tested cell lines, MEL caused the contraction of cell bodies and cell nuclei. Treatment of GN and GO alone did not cause visible changes in structure, while the examined complexes caused significant cell degradation. In the case of the MDA-MB-231 and MCF-7 lines, there was a noticeable reduction in the number of viable cells, because it was impossible to find an area on the preparation that contained a comparable number of cells in one field of view at the same magnification compared to the control group. In the cells that remain, mainly cell nuclei were visible, while the fluorescence from labeled actin was much lower than in the control group, which may be due to damage to the cytoskeleton.

One of the differences between cancer cells and healthy cells is intracellular pH value. Healthy cells strive to achieve a pHi close to neutral using membrane proteins that transport ions between the intercellular environment and the interior of the cell [63,64,65]. Cancer cells have a higher pHi, usually ~7.3–7.6, compared with normal cells at ~7.2. The increased pHi is maintained by the increased activity of ion membrane transporters, such as MCT or NHE1, responsible for the regulation of pHi, despite the increased metabolism of cancer cells and greater production of metabolic acids. The pHi change is correlated with many cellular processes, one of which is migration. Many studies showed that increased pHi level is crucial during cell migration. This phenomenon is necessary to change the actin fibers in the cells. It has been shown that the increase in pHi is caused by the activity of ion channels, which is related to the phenotype of cellular invasion and enables the migration of breast cancer cells [66]. On the other hand, the increase in pHi is caused by a reduction of CO_2_ production correlated with oxidative phosphorylation and a shorter tricarboxylic acid cycle [67]. In our study, a significant increase in fluorescence was observed for the groups treated with M and MGN for all cell lines, which translates into a decrease in pHi. In the case of M, this may be due to the disruption in the cell membrane and a lowering of the pH caused by the naturally lower extracellular pH of the tumor cells. Intracellular pH is also lowered in the event of the activation of apoptosis, which may have occurred after the treatment of cells with the MGN complex.

Changes in pHi, and, thus, the activity of ion channels, are also associated with a change in the potential of the cell membrane. The potential of the cell membrane is created by ion channels and transporters, which have specific and selective permeability. In the case of neoplastic cells, they are characterized by a greater polarization of the membrane compared with healthy cells [68,69]. In the case of cancers, polarization of cell membranes is associated with the ability to metastasize; therefore, influencing this cellular factor may limit the migration of neoplastic cells in the body [70]. In the conducted study, a significant decrease was observed in the potential of the cell membrane caused by MEL itself and the MGN complex in all cell lines, while MGO caused a significant decrease in membrane polarization for lines MDA-MB-231 and HFFF2. According to the literature, it may reduce the metastatic capacity of the MDA-MB-231 and MCF-7 lines and increase the toxic effect for all lines.

It is evident that MEL degrades cells by creating pores in the membrane, while carbon nanoparticles are taken into cells by internalization [40]. In the conducted study, it was checked whether the MEL complexes with GN and GO are also taken up by cells, just like the nanoparticles themselves. For this purpose, a viability test was performed using the following cellular transport blockers: Pitstop-2, colchicine, cytochalasin D, genistein, phenylorsine oxide, chlorpromazine, and dynasore hydrate.

Pitstop-2 is a clathrin inhibitor of clathrin-mediated endocytosis (CME) [71]. Colchicine is a drug used in patients with gout [72]. At the cellular level, it inhibits microtubule polymerization by binding to tubulin, which reduces microtubule-based transport [73]. Another transport inhibitor used was cytochalasin D, which is a potent inhibitor of actin polymerization [74]. It is a fungus-derived compound with the ability to bind the barbed ends of actin filament and block phago- and micropinocytosis. Genistein acts as a protein tyrosine kinase inhibitor derived from soya beans. It can be used as an inhibitor of caveolin-dependent endocytosis (CDE) [75]. Phenylarsine oxide is another cellular transport blocker and inhibits protein tyrosine phosphatases, but its mechanism remains unknown. In research, it is used as an inhibitor of phagocytosis and micropinocytosis [76,77]. Chlorpromazine is another inhibitor of CME, which inhibits clathrin disassembly and receptor recycling to the plasma membrane [78]. Dynasore hydrate has an inhibitory effect on dynamin-dependent endocytosis by inhibiting the GTPase activity of dynamin1 and dynamin2 [79,80].

The results showed that for the MDA-MB-231 cell line, the toxic effect of the MGN complex was blocked only with genistein, which means that it is taken up via CDE. This can be concluded because the use of a compound that prevents the cells from this type of transport did not decrease the viability of the cells after treatment with MGN, as in the case of other blockers and no-blocker controls. The lack of change in viability after the use of genistein was also observed for MGO, which proves that both complexes are taken up by the same mechanism to MDA-MB-231 cells. For the MCF-7 cell line, a significant decrease in viability was observed compared to the control group with the use of all transport blockers, except phenylarsine oxide, which means that the MGO complex can be taken up into the cell by phagocytosis. Interestingly, for MGO in MCF-7 cells, a decrease in viability by about 30% compared to control was observed after exposure to chlorpromazine, which may mean that this complex may be partially taken up by CME. For the MCF-7 line, a decrease in viability after treatment with MGN was observed in all groups, which may mean that this complex can be delivered by various types of transport. Interestingly, for MDA-MD-231 cells, a significant decrease in viability after treatment with MGO was not observed after exposure to phenylarsine oxide, colchicine, cytochalasin D, and genistein, which is associated with the ability of MDA-MB-231 cells to take up MGO by phagocytosis, microtubule-dependent transport, and CDE. For the HFFF2 healthy cell line, no decrease in viability was observed in the MGO-treated group after the use of phenylarsine oxide, cytochalasin D, and colchicine; thus, the results obtained suggest that MGO is transported into HFFF2 cells by phagocytosis and microtubule-based transport. For all tested lines, the toxic effect of MEL was not eliminated by the applied blockers, which confirms the known mechanism of its toxic action, that is, through the formation of pores in the outer membrane of the cell. The MGN complex showed a toxic effect on HFFF2 cells in all research groups, which may be because it can be taken up by various types of transport.

Analysis of the protein level showed an increase in ALCAM protein expression in MDA-MB-321 cells after exposure to the MGN complex and, interestingly, a decrease in expression in MCF-7 cells treated with MGN and MGO complexes. Differences in the response of this protein expression between lines may result from the structure of their cell membranes because the MDA-MB-231 line is a triple-negative line, while MCF-7 has estrogen, progesterone, and glucocorticoid receptors. ALCAM is a membranous protein and takes part in cell migration and adhesion; also, it is usually expressed in breast cancer cells. Lowering ALCAM expression in this type of cancer is associated with poor prognosis for patients [81]. It has been shown that patients with bone metastases have the lowest expression of this protein and that is characteristic of patients with undesirable features of the tumor and unfavorable clinical outcomes [82]. The increase in expression of this membranous protein has a lowering effect on adherent ability and metastasis [83]. Further, ALCAM is known as a protector against apoptosis and autophagy for breast cancer cells [84]. Our study showed that the expression of this receptor was increased in cells of the triple-negative MDA-MB-231 line in response to exposure to the MGN complex, while the reverse effect was observed in the MCF-7 line where expression decreased in the groups treated with MNG and MGO.

Lipocalin-2 is a protein involved in the transport of hydrophobic and small particles, such as steroids and hormones [85,86]. In recent years, this protein expression has been studied in various diseases, and its impaired expression has been attributed, inter alia, to damage to the liver and kidneys and to human neoplastic diseases [87,88]. Its higher expression has been shown in neoplastic cells of epithelial origin, such as ovarian, lung, or breast cancer. This protein has become a biomarker for some types of cancer and an index of malignancy. It was reported that NGAL has increased expression in carcinoma tissues, urine, and sera of patients with breast cancer. It has been shown that NGAL actively promotes breast cancer metastasis by inducing VEGF production, EMT, angiogenesis, and cell invasion and migration through multiple signaling pathways, including PI3K/AKT/NF-Κb [87,89,90,91,92,93,94,95]. The decrease in expression of NGAL was observed in MDA-MB-231 cells treated with MGN and MGO complexes, which indicates that the tested complexes may reduce the ability to metastasize triple-negative breast cancer. These results increase the likelihood of increasing metastatic capacity along with the results for NGAL protein expression.

uPAR is the protein in the cell membrane known as a urokinase plasminogen activator surface receptor or CD87 [96]. The function of this protein on the surface of cancer cells is coupled with the lytic effect of uPA, which allows for the degradation of the extracellular matrix (ECM) and cell migration [97]. Protein has the ability to regulate both cell proliferation and migration, contributing to tumor progression. Currently, uPAR is one of the prognostic indicators of neoplastic diseases and its high level in the serum of patients does not favor a good prognosis [98]. Analysis of the protein level showed an increase in this protein in MDA-MB-231 cells treated with both tested complexes, which, combined with previous toxicity studies, may indicate a defense reaction of cells and, translated into the action in the body, a desire to “escape” from the toxic factor [97,99,100,101,102,103]. 

The 4-1BB receptor is expressed mainly in the cells of the immune system and belongs to the tumor necrosis factor (TNF) receptor family but has also been shown to be expressed in tumor cells [104,105]. In cancer, it has been shown that anti-4-1BB-agonistic IgG can reduce the growth of tumors [106]. In contrast, the 4-1BB protein is involved in the regulation of monocytes/macrophages in the tumor microenvironment [107]. Moreover, it is a factor that promotes the metastasis of breast cancer to the bone [108]. In the conducted analysis, an increase in the expression of this protein was observed for the MDA-MB-231 line in cells treated with MEL. This may be related to the cell’s defenses and an attempt to alter the environment that is toxic to the cells, similar to a uPAR protein. This is consistent with the previous finding that MEL is toxic to this cell line.

The TRIAL-R3 receptor belongs to the TNF receptor superfamily [109]. It is composed of the extracellular TRIAL binding domain and the transmembrane domain; however, it does not contain the cytoplasmic death domain [110,111]. It is not responsible for the induction of apoptosis; it is, indeed, believed that it competes with other receptors to protect cells from induction of this TRIAL-induced cell death pathway. It is found in both healthy cells and some cancer cells. An increase in expression was observed for the MCF-7 line after exposure to MEL. This may be related to blocking the induction of apoptosis, the mechanism of cell death. 

Dtk protein, also known as TYRO3 is a tyrosine-protein kinase receptor [112]. It participates in the transmission of signals from the microenvironment to the cytoplasm, inter alia, by binding to TULP1 and GAS6 [113]. It is also involved in activating the AKT survival pathway, which is also associated with cell proliferation [114]. In the MCF-7 cell line, the increase in expression of this receptor after MGN treatment was observed, which may mean activation of the AKT pathway. It is known that AKT belongs to one of the activated kinases on the human list. Activating this kinase can promote proliferation and increase their defense against the activation of apoptosis. Thus, the cells could begin their defense processes after being exposed to the MGN complex. This may be the result of the high EE% and high MEL content in the complex with GN, which enters the cell together with the nanoparticle.

In a healthy organism, the MICA protein shows low expression in healthy cells, such as the intestinal epithelium, monocytes, and fibroblasts. However, in the event of a cell imbalance, it can be produced by cells as a response to stress. The increase in MICA levels is observed during viral infections, heat shock, and DNA damage. The protein interacts with the group of natural 2D killers (NKG2D), which activates the cytolytic effect of T lymphocytes and NK cells on neoplastic cells of epithelial origin [115,116]. However, cancer cells have developed a defense mechanism of shedding MICA from the cell surface, making it difficult for the immune system to recognize it as dangerous. In the case of this receptor, a decrease in protein expression and level was observed in both tested cell lines. However, the difference is in the study groups where a decrease was observed relative to the control group. For the MDA-MB-231 line, the decrease in expression was noted after treatment with MEL, while in MCF-7, for both tested MEL complexes with nanoparticles, a decrease was observed. This may indicate DNA damage caused by MEL in the cell line MDA-MB-231 and MEL derived from complexes in cells of the MCF-7 line.

To summarize, the MDA-MB-231 and MCF-7 lines, which differ in the presence of estrogen receptors on the cell surface, react differently to the studied MEL complexes and carbon nanoparticles (Figure 14). The differences were shown by the analysis of the expression of membrane receptor proteins and the potential of the cell membrane. In turn, the MGN complex showed a greater reduction in viability than the MEL itself, which may be related to the high entrapment of MEL with GN and its reduced interaction with the outer membrane and increased interaction with intracellular structures. The use of nanoparticles as carriers did not reduce the toxicity to healthy cells, but these were in vitro studies, where one cell line was grown in one bottle, so in vivo studies should be carried out, where healthy and neoplastic cells are in contact with each other.

## 5. Conclusions

The studied cancer lines react differently to the tested complexes of MEL with GN and GO, which may be the result of the presence or absence of estrogen receptors on the cell surface. For this reason, we demonstrated the effect of these complexes by examining the change in cell membrane potential, intracellular pH, the type of transport by which the tested complexes are delivered to the cell interior, and the analysis of receptor protein expression. However, despite the promising results for the tested complexes, it is necessary to focus on their toxicity to healthy tissues surrounding tumors in the body and check whether topical application is sufficient to limit the toxic effects of the complexes on non-cancer cells.

## Figures and Tables

**Figure 1 jfb-13-00278-f001:**
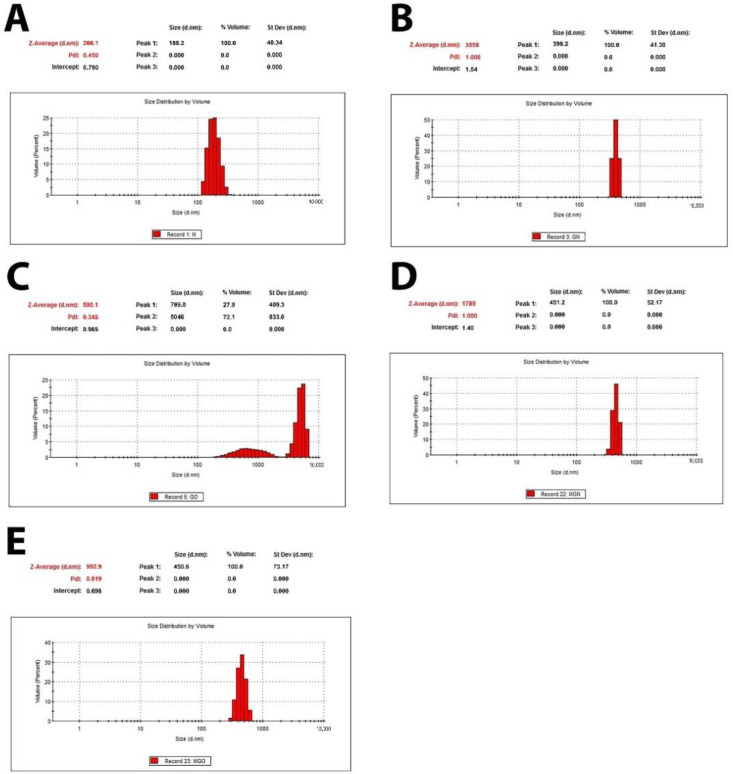
Size distribution of MEL, GN, GO, and complexes. (**A**)—MEL; (**B**)—GN; (**C**)—GO; (**D**)—MGN; (**E**)—MGO.

**Figure 2 jfb-13-00278-f002:**
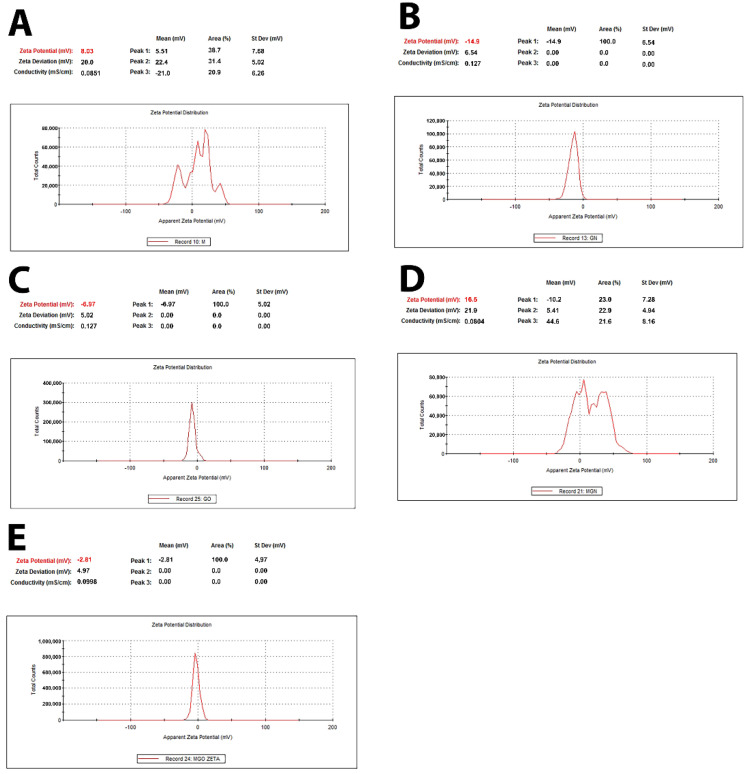
Zeta potential of MEL, GN, GO, and complexes. (**A**)—MEL; (**B**)—GN; (**C**)—GO; (**D**)—MGN; (**E**)—MGO.

**Figure 3 jfb-13-00278-f003:**
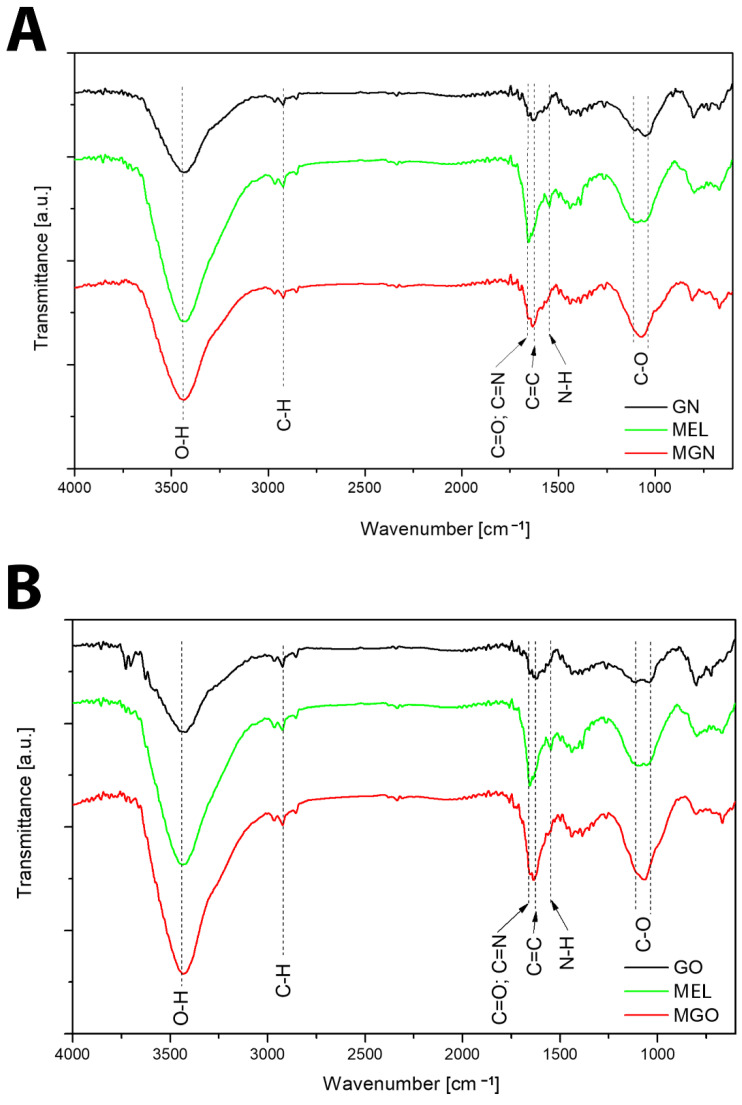
FT-IR spectra. (**A**)—GN, MEL, MGN; (**B**)—GO, MEL, MGO.

**Figure 4 jfb-13-00278-f004:**
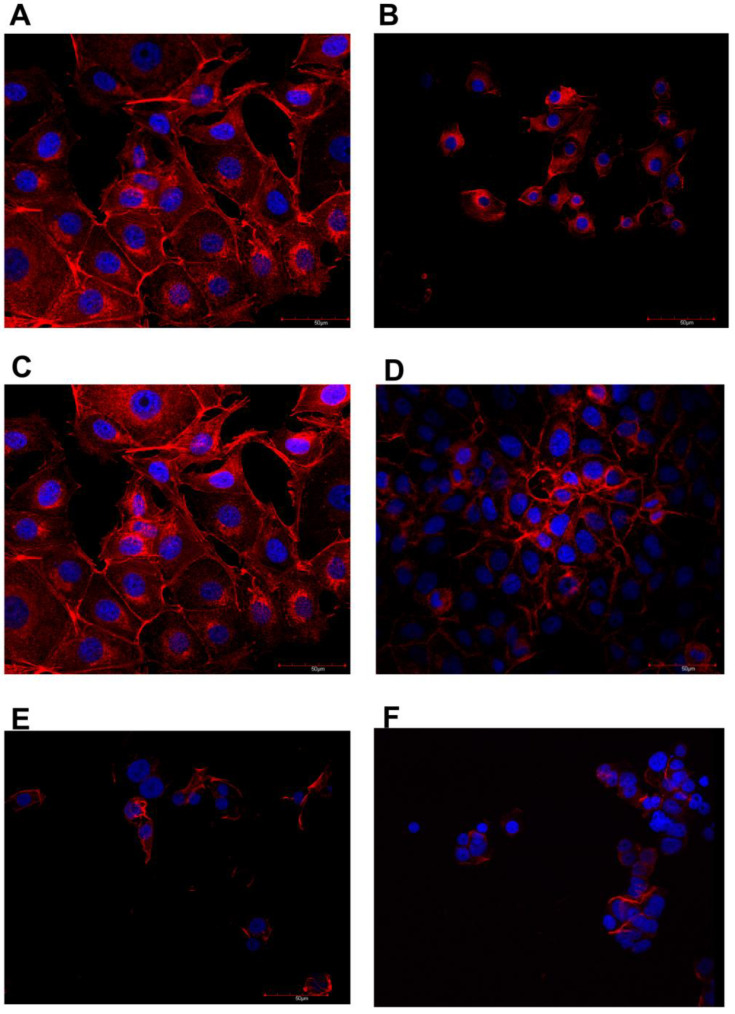
Immunofluorescence images of MCF-7 line cells stained for actin filaments (red) and nuclei (blue). (**A**)—control, (**B**)—MEL-treated group, (**C**)—GN-treated group, (**D**)—GO-treated group, (**E**)—MGN-treated group, and (**F**)—MGO-treated group.

**Figure 5 jfb-13-00278-f005:**
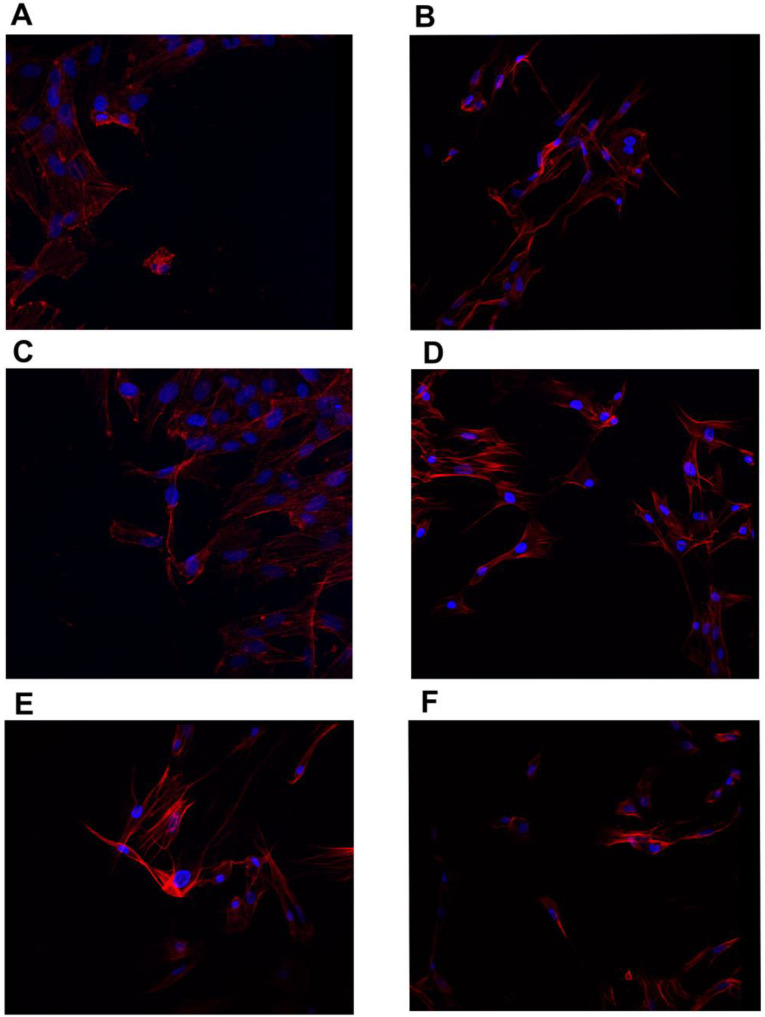
Immunofluorescence images of HFFF2 line cells stained for actin filaments (red) and nuclei (blue). (**A**)—control, (**B**)—MEL-treated group, (**C**)—GN-treated group, (**D**)—GO-treated group, (**E**)—MGN-treated group, and (**F**)—MGO-treated group.

**Figure 6 jfb-13-00278-f006:**
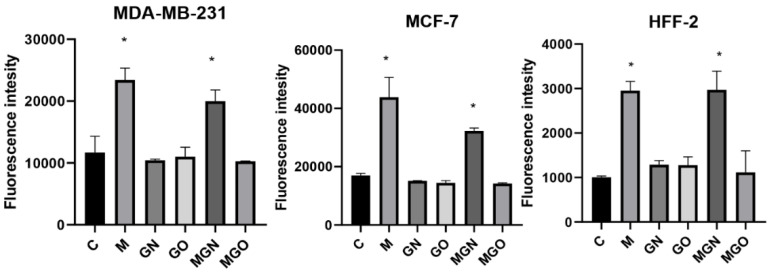
Intracellular pH in cell lines after treatment with MEL, nanoparticles, and complexes. C signifies the control samples and the results are mean values ± standard deviation. “*” indicates statistically significant differences in comparison to the control (*p*-value ≤ 0.05).

**Figure 7 jfb-13-00278-f007:**
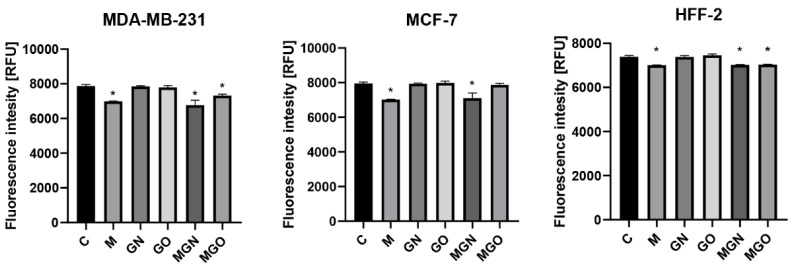
Potential of cell membrane in cell lines after treatment with MEL, nanoparticles, and complexes. C signifies the control samples and the results are mean values ± standard deviation. “*” indicates statistically significant differences in comparison to the control (*p*-value ≤ 0.05).

**Figure 8 jfb-13-00278-f008:**
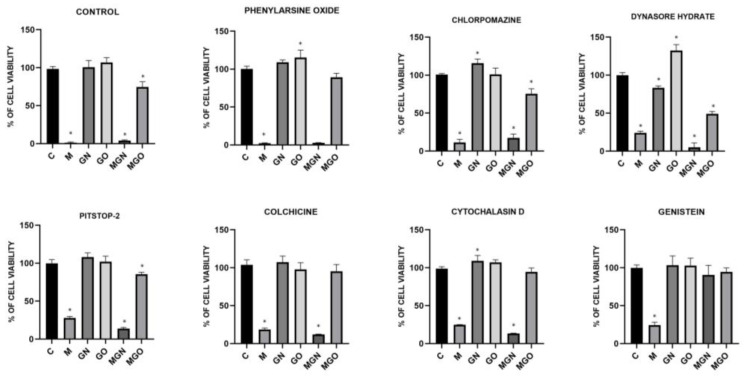
Effect of transport inhibitors on the internalization of M, GN, GO, MGN, and MGO into MDA-MB-231 cells. C signifies the control samples and the results are mean values ± standard deviation. “*” indicates statistically significant differences in comparison to the control (*p*-value ≤ 0.05).

**Figure 9 jfb-13-00278-f009:**
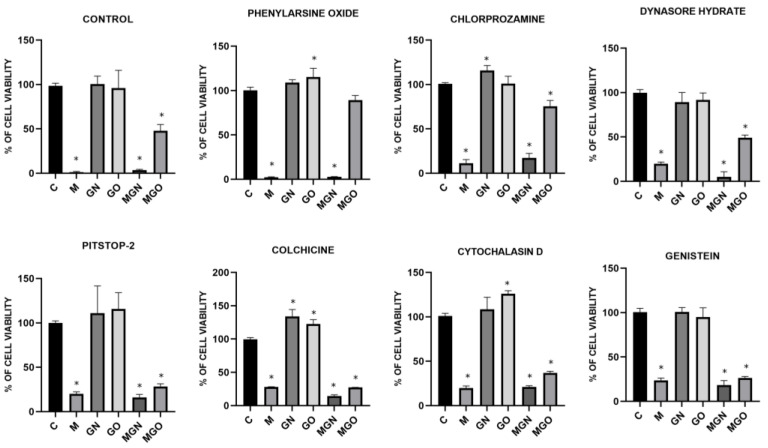
Effect of transport inhibitors on the internalization of M, GN, GO, MGN, and MGO into MCF-7 cells. C signifies the control samples and the results are mean values ± standard deviation. “*” indicates statistically significant differences in comparison to the control (*p*-value ≤ 0.05).

**Figure 10 jfb-13-00278-f010:**
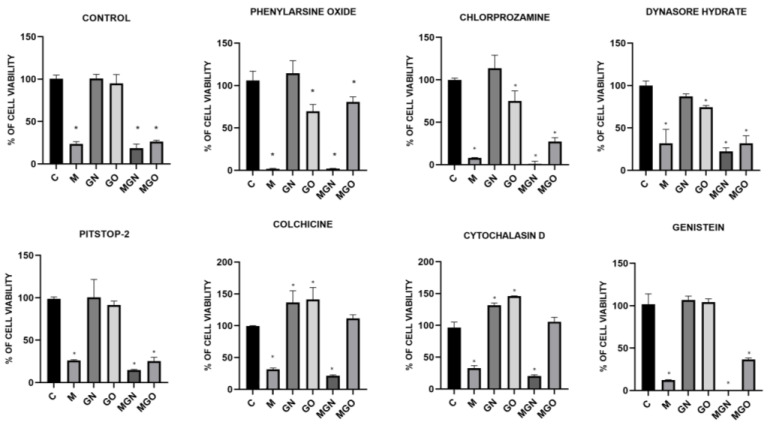
Effect of transport inhibitors on the internalization of M, GN, GO, MGN, and MGO into HFFF2 cells. C signifies the control samples and the results are mean values ± standard deviation. “*” indicates statistically significant differences in comparison to the control (*p*-value ≤ 0.05).

**Figure 11 jfb-13-00278-f011:**
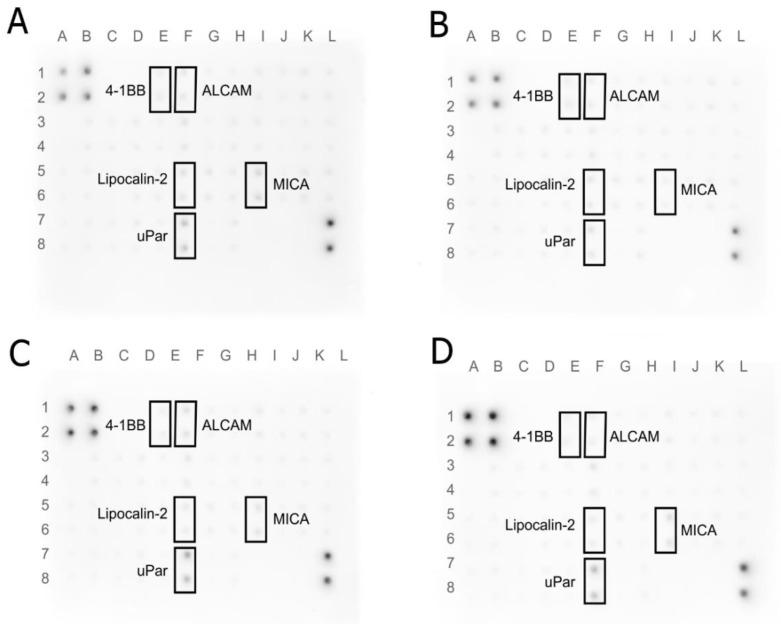
Antibody array analysis of human cell membrane receptor (original drafts) in MDA-MB-231 cells. (**A**)—control group, (**B**)—MEL-treated group, (**C**)—MGN-treated group, and (**D**)—MGO-treated group. Results were normalized and compared to a dots control sample.

**Figure 12 jfb-13-00278-f012:**
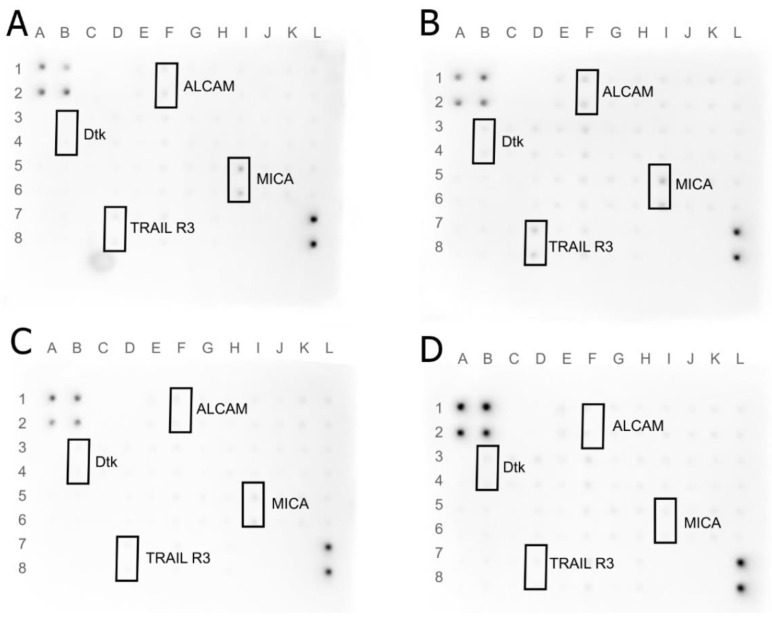
Antibody array analysis of human cell membrane receptor (original drafts) in MCF-7 cells. (**A**)—control group, (**B**)—MEL-treated group, (**C**)—MGN-treated group, and (**D**)—MGO-treated group. Results were normalized and compared to a dots control sample.

**Figure 13 jfb-13-00278-f013:**
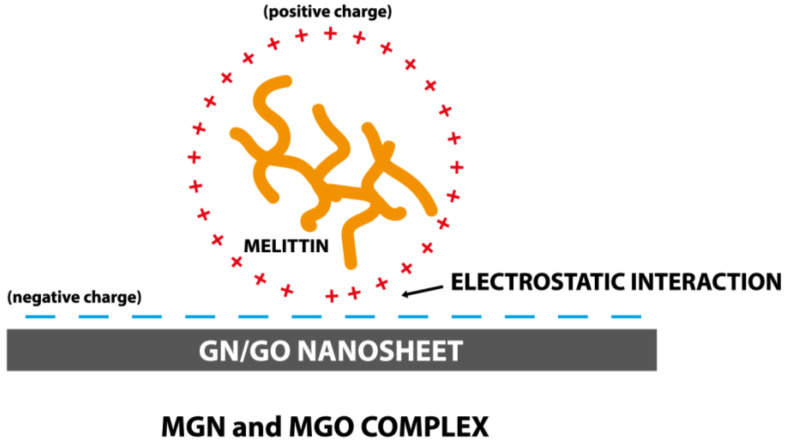
Scheme of MGN and MGO complexes forming.

**Figure 14 jfb-13-00278-f014:**
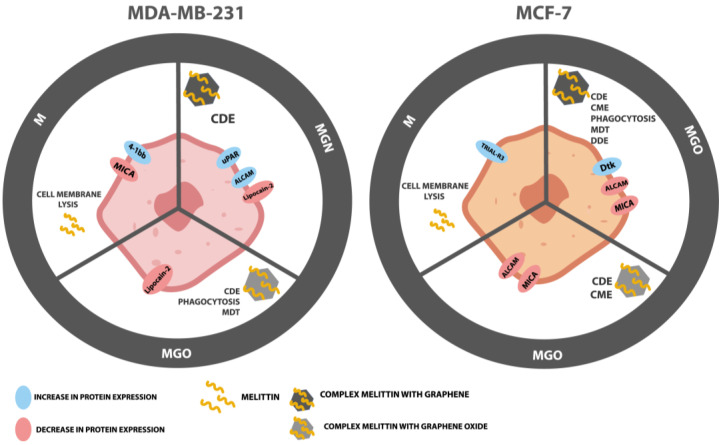
Mechanism of action of MEL, MGN, and MGO on MDA-MB-231 and MCF-7 cells; CDE—caveolin-dependent endocytosis; CME—clathrin-mediated endocytosis, MDT—microtubule-dependent transport, DDE—dynamin-dependent endocytosis.

## Data Availability

The data presented in this study are available on reasonable request from the first author.

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
