# Peer review of "Delivery of Melittin as a Lytic Agent via Graphene Nanoparticles as Carriers to Breast Cancer Cells"

_jfb, 2022, doi:10.3390/jfb13040278_

Round 1
Reviewer 1 Report
The authors of the study investigated the mechanism of delivering melittin as a lytic agent by graphene and graphene oxide nanoparticles as carriers to breast cancer cell lines MDA-MB-231 and MCF-7. This study can be published after following corrections
1. The manuscript contains several typographical errors 8ie. Name of chemical compounds. The position of peak should be given as cm-1 superscript) and grammatical. The authors should carefully read the article and improve the quality of the MS file.
2. Figure 1 must be revised and the corresponding peaks (ie. 1650 cm-1 and other peaks) must be labelled on the peaks. The others should also present Figure 1 and Figure 2 under Figure 1 as Figure 1a and Figure 1b.
3. The legends of figures need more explanation. Figure 6- Figure 10 needs more statistical explanation ie. P<0.005 or P<0.05. What does * mean in terms of p value?
4. Following sentences must be added either into introduction or discussion to comprehensively introduce the topic.
“For decades, natural ingredients have been employed for therapeutic purposes. Their beneficial effects in easing and treating a variety of illnesses, including cancer. Breast cancer is the second-most common cause of cancer-related death for American women and the primary cause of cancer death for women globally. Breast cancer risk can be determined using risk assessment methods, and patients who are at a high risk may be candidates for drugs that lower their risk (1-3).
1. Özbolat, S. N., & Ayna, A. (2021). Chrysin suppresses HT-29 cell death induced by diclofenac through apoptosis and oxidative damage. Nutrition and Cancer, 73(8), 1419-1428.
2. Trayes, K. P., & Cokenakes, S. E. (2021). Breast cancer treatment. American Family Physician, 104(2), 171-178.
Author Response
Dear reviewer,
Thank you for the valuable comments.
The manuscript is modified in accordance with your comments. All new sentences have been marked with "Track Changes".

Reviewer 2 Report
In the present study, Daniluk and co-authors present an interesting research paper about the delivery of melittin through graphene nanoparticle approach for breast cancer therapy. The work is good for drug delivery applications. This is an excellent work - well organized, thoroughly reviewed, well written, and covers the all the required work that needs to be done. The manuscript is very well presented and enjoyable to read.
There are a few minor issues that should be taken into consideration, as indicated in the following:
1. Please modify the abstract with respect to numerical values of obtained data e.g., encapsulation efficiency, FT-IR interaction studies, pH etc etc.
2. The manuscript should clearly state the novelty of work in the introduction.
3. The title should be changed as in the present form its confusing, I think you should focus on delivery of Melittin instead of mechanism.
4. I suggest, authors should also go for In vitro drug release studies to check the amount of Melittin release in the breast cancer related medium under specified conditions. So that we can get an idea of how much amount of melittin is release in a given time.
5. Kindly reduce the number of abbreviations in the whole manuscript.
6. Please add some latest articles of year 2021 and 2022.
7. References needs to be checked as per journals requirements.
8. A conclusion section should be there to summarise your findings.
9. Moreover, also the addition of very recent papers is desirable to improve the introduction part.
· Asmita Deka Dey, Ashkan Bigham, Yasaman Esmaeili, Milad Ashrafizadeh, Farnaz Dabbagh Moghaddam, Shing Cheng Tan, Satar Yousefiasl, Saurav Sharma, Ana Cláudia Paiva-Santos, Aziz Maleki, Navid Rabiee, Alan Prem Kumar, Vijay Kumar Thakur, Gorka Orive, Esmaeel Sharifi, Arun Kumar, Pooyan Makvandi. Dendrimers as nanoscale vectors: Unlocking the bars of cancer therapy, Seminars in Cancer Biology, 2022

Author Response

(The authors gave the same response as above.)

Reviewer 3 Report
In the presented manuscript, titled “Mechanism of delivering melittin as a lytic agent by graphene nanoparticles as carriers to breast cancer cells”, Daniluk et al. report the study of the properties of melittin-loaded graphene or graphene oxide, and the delivery mechanism is investigated using biological assays. My main concerns about this work rely on the poor description and characterization of the reported graphene/melittin complex. Moreover, I believe that either breast cancer active targeting or in vivo studies are required to prove the biocompatibility and non-toxicity of the proposed complex system. Therefore, I don’t think this work meets the minimum requirements for publication on Journal of Functional Biomaterials (IF = 4.901), and I recommend the Editor for the rejection of the presented manuscript in its current form.
In particular, few issues need to be addressed before taking this work into further consideration:
1) It is not clear from the introduction why would it be required to internalize melittin in breast cancer cells. In lines 51-53, the authors state that “The main mechanism of MEL is membrane action, which contributes to the increased interest in its role in clinical application in anticancer research.” Why did the authors aim at using GN or GO to improve MEL internalization if its main action is on the cell membrane? Please provide a rationale.
2) All acronyms should be expanded the first time they are used in the manuscript text. In particular, ND, MGO and MGN.
3) The nature of the interaction between MEL and the nanomaterials should be discussed. Is it electrostatic? If yes, please provide Zpot data showing opposite surface charge of the two components. During the preparation of the complex, the two components are simply mixed with no purification performed to remove unbound MEL or nanomaterial. The interaction between the two components must be proved, because in the current form the manuscript describes a mere physical mixture of MEL and graphene, and rather that speaking of a nano-complex the authors should be speaking of a synergistic effect the two components. Moreover, a Figure should be added representing schematically how the complex is formed.
4) The synthetized graphene/melittin complex should be characterized in details in order to allow for an appropriate interpretation of the experimental results. For example, the effect of the complexation n the nanosystem morphology should be evaluated by TEM and/or DLS.
5) Details of the quantification of MEL using NanoDrop should be provided. For example, it should be specified whether the standards were prepared using MEL or other proteins, in which range of concentration they were prepared and at which wavelength the detection of MEL was performed.
6) Quality of IR spectra should be improved, for example by performing baseline subtraction for both figure 1 and 2.
7) Text in Figure 13 cannot be read.
8) In vivo experiments or active targeting approaches should be investigated to prove the effectiveness of the proposed approach in real case scenarios.
Author Response

(The authors gave the same response as above.)

Round 2
Reviewer 3 Report
Most of the highlighted issues have been addressed.